METHODS

# Denoising diffusion probabilistic models for generation of realistic fully-annotated microscopy image datasets

**Dennis Eschweiler**⦿*, **Rüveyda Yilmaz**⦿, **Matisse Baumann, Ina Laube, Rijo Roy, Abin Jose, Daniel Brückner, Johannes Stegmaier**⦿*

RWTH Aachen University, Institute of Imaging and Computer Vision, Aachen, Germany

* dennis.eschweiler@lfb.rwth-aachen.de (DE); johannes.stegmaier@lfb.rwth-aachen.de (JS)

## Abstract

Recent advances in computer vision have led to significant progress in the generation of realistic image data, with denoising diffusion probabilistic models proving to be a particularly effective method. In this study, we demonstrate that diffusion models can effectively generate fully-annotated microscopy image data sets through an unsupervised and intuitive approach, using rough sketches of desired structures as the starting point. The proposed pipeline helps to reduce the reliance on manual annotations when training deep learning-based segmentation approaches and enables the segmentation of diverse datasets without the need for human annotations. We demonstrate that segmentation models trained with a small set of synthetic image data reach accuracy levels comparable to those of generalist models trained with a large and diverse collection of manually annotated image data, thereby offering a streamlined and specialized application of segmentation models.

**Data Availability Statement:** All code written in support of this publication is publicly available at https://github.com/stegmaierj/

## Author summary

Modern generative techniques have unlocked the potential to create realistic image data of high quality, prompting the possibility of substituting real image data in segmentation training workflows. Our study highlights the capacity of denoising diffusion probabilistic models to generate high-quality microscopy image data. With adjustments to the generation process, these models can produce realistic fully-annotated image datasets through an intuitive and unsupervised approach. The parameters of the generative pipeline undergo optimization through various evaluations, resulting in synthetic image data that exhibits high PSNR scores. Our practical experiments encompass multiple scenarios, including manual annotations, initial segmentations, and simulations as starting points, demonstrating the versatility of our approach. Importantly, we compare the performance of segmentation models trained on a limited set of synthetic image data with those trained on a vast and diverse collection of manually annotated data, demonstrating the potential of our pipeline to alleviate the reliance on extensive manually annotated datasets. Our approach lays the groundwork for similar applications, thereby promoting the much-

DiffusionModelsForImageSynthesis and simulated data is available at https://osf.io/dnp65.

**Funding:** This work was funded by the German Research Foundation DFG with the grants STE2802/2-1 (DE) and STE2802/1-1 (IL). The funders had no role in study design, data collection and analysis, decision to publish, or preparation of the manuscript.

needed availability of publicly accessible fully-annotated image datasets and advancing the goal of annotation-free segmentation.

This is a *PLOS Computational Biology* Methods paper.

## Introduction

Enabling automated segmentation of cells in fluorescence microscopy image data is a crucial step in supporting biomedical experts in conducting a large variety of experiments [1, 2]. This variety in experimental settings is mirrored to the image data appearances, posing a challenge for segmentation approaches trained with the generally scarce variety of annotated image data. To overcome this challenge, costly and tedious human annotations have to be acquired, causing a bottleneck in realizing the full potential of learning-based approaches and restricting their application in practice. Annotation efforts are reduced by automated data augmentation approaches [3–5] and tweaked segmentation pipelines [6, 7], which help to ease the challenge, but still demand a small set of fully-annotated image data as a basis. Alternatively, automated simulation approaches replicate desired characteristics of cellular structures in arbitrary amounts of image data [8–13] and ideally serve as a way to entirely replace human annotation.

Recently, denoising diffusion probabilistic models (DDPM) [14] have shown great potential in generating realistic image data [15, 16], while neither requiring annotated training data, nor adversarial training concepts, as opposed to commonly used generative adversarial networks (GAN) [9, 10, 13]. GANs often present challenges during training and can be susceptible to mode collapse, which results in non-convergent training behavior [17]. In contrast, DDPMs offer enhanced diversity due to their likelihood-based principle, making them favorable in terms of data quality. However, for the purpose of generating fully-annotated datasets suitable for training segmentation methods, generative approaches must be strictly conditioned. They should have the capability to convert annotation masks into realistic image data while precisely preserving their structural specifications to maintain the faithfulness of the corresponding annotations. This necessitates the modification of the regular DDPM data generation process.

In this study, we explore the potential of diffusion models for generating microscopy image data across various organisms. We present a novel pipeline that utilizes DDPMs for intuitive data generation, employing rough sketches of desired structures as a basis, followed by training segmentation algorithms with the generated data (Fig 1a). To evaluate the pipeline's effectiveness for different scenarios, we conduct diverse experiments employing manual annotations, initial erroneous segmentation outputs, or simulated data as the basis for sketches. The practical applicability of our approach is demonstrated by obtaining segmentation results without the need for human annotations. These results are then compared with outcomes from generalist segmentation models trained on a large and diverse dataset of manually annotated images. Furthermore, we provide public access to our code and the fully-synthetic image datasets, aiming to provide the tools for data generation and enhance the accessibility of fully-annotated image datasets for the broader research community.

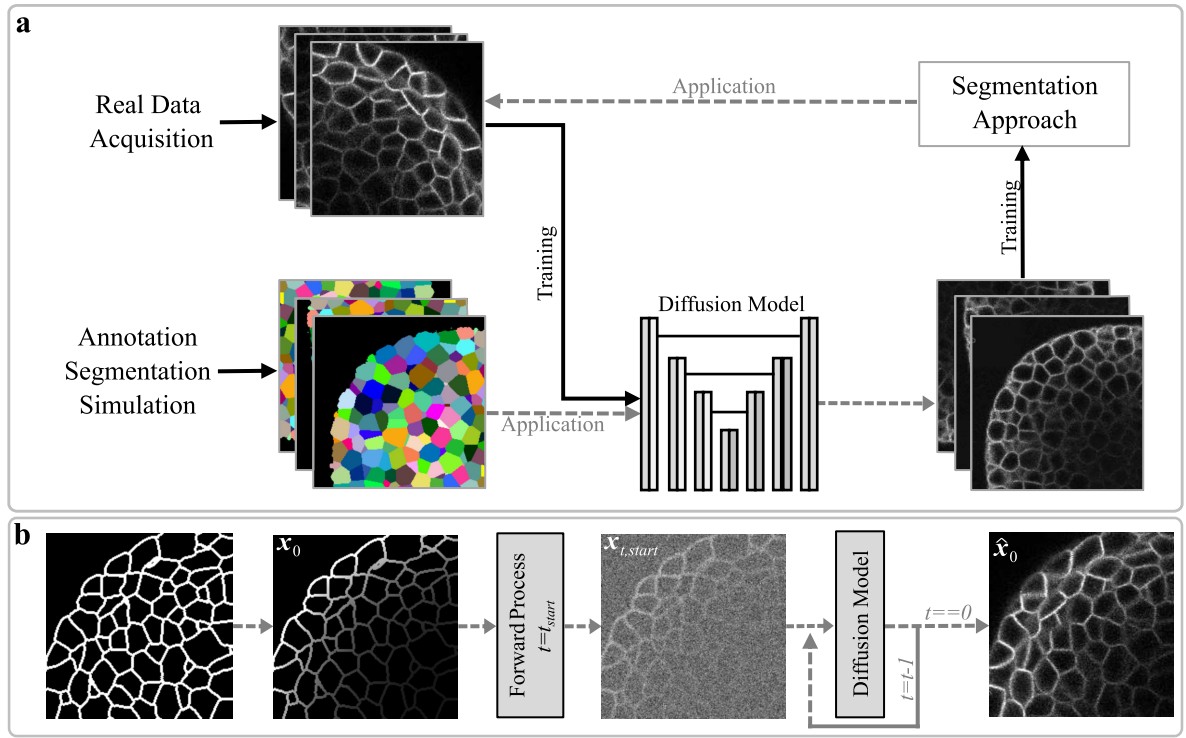

**Fig 1. Pipeline overview.** (a) The whole pipeline involves training a diffusion model on real image data and applying it to obtained structures to generate fully-annotated image datasets, which are then used to train models that segment the real data. (b) During application of DDPMs, annotations are automatically turned into coarse sketches for a subsequent application of the forward process, to achieve a realistic generation of the corresponding image data.

## Results and discussion

The utilized datasets show either cellular membranes or cell nuclei obtained from 2D(+t) and 3D fluorescence microscopy experiments. Among those are a publicly available 3D fluorescence microscopy image data set showing the meristem of *A. thaliana* [18] and corresponding manually corrected annotation masks. Due to the availability of full annotations, this dataset was used for optimization experiments and detailed analyses. Further datasets include 3D nuclei of developing *C. elegans* [19, 20], 3D nuclei of developing *T. castaneum* embryos [19], a multi-channel 3D dataset showing fluorescently labeled cell membranes and nuclei in two different zebrafish embryos [21], 2D mouse stem cells [19, 22], 2D HeLa cells [19, 23] and a 2D+t dataset showing temporal mitotic progression of HeLa cells [24].

In the concept of denoising diffusion probabilistic models (DDPM), a forward process iteratively transforms an image into pure noise by the incremental addition of small portions of noise to an initial image $\mathbf{x_0}$. The generation of realistic image data is performed by a learned backward process and typically starts from this pure noise state at the last timestep T (Fig 1b). However, starting from pure noise $\mathbf{x}_T$ does not allow to generate fully-annotated datasets due to the lack of control over the generated structures and the absence of corresponding annotations. To address those issues, two adaptions are made to the application of DDPMs. First, the backward process is initiated at an early timestep $t_{start} <$ T, ensuring that a significant portion of structural indications in $\mathbf{x}_{t,start}$ is not yet fully obscured by noise and can be preserved throughout the generation of realistic image data [25]. Second, to allow for an intuitive modelling and control over cellular structures within the generated image data, sketches replace real

image data as a starting point for the forward process generating $\mathbf{x}_{t,start}$. These sketches provide indications of cell positions, shapes, and coarse structural characteristics, specifying a brief outline of the desired scene to be generated. As the learned backward process was solely trained on real image data, the subsequent application to $\mathbf{x}_{t,start}$ results in the generation of corresponding realistic image data (Fig 1b). Ultimately, with the known cell outline and positioning within the sketches, the pipeline is able to generate realistic fully-annotated image data in a automated and unsupervised manner. However, in order to maintain high realism during the backward process, it is crucial for the noisy samples $\mathbf{x}_{t,start}$ originating from the sketch domain $\mathcal{M}$ to contain enough noise to exhibit data distributions that closely resemble those originating from the real image domain $\mathcal{I}$. Simultaneously, it is essential that structural information given in the sketches is preserved despite the introduction of noise in $\mathbf{x}_{t,start}$. This poses an optimization challenge in determining the optimal value for the parameter $t_{start}$.

## Pipeline optimization

Optimizing the point of initializing the backward process $t_{start}$ requires balancing the generation of fine-grained details with the preservation of structural correlation to sketches. To generate fine details, a substantial noise content from later stages of the forward process is necessary, but this can compromise the preservation of structural indications, which requires stopping the forward process as early as possible. The details of this trade-off were evaluated using a publicly available fully-annotated 3D microscopy image dataset [18], which provides manually corrected annotations that enabled a precise assessment of various aspects of the proposed pipeline. During examination of this trade-off, we found that applying a Gaussian smoothing with standard deviation $\sigma$ to sketches before applying the forward process helped to prevent unnaturally sharp edges and reach similar data distributions earlier. To analyze the forward process, both $\mathbf{x}_{t,start}^{\mathcal{M}}$ and $\mathbf{x}_{t,start}^{\mathcal{I}}$ were generated, and their similarity was measured by constructing data distribution histograms (Fig 2a) and calculating the Bhattacharyya distance and Kullback-Leibler divergence as shown in Table 1. Additionally, to assess the learned backward process, all noisy samples were used to generate realistic $\hat{\mathbf{x}}_0$, which were quantitatively evaluated using the peak signal-to-noise ratio (PSNR) for textural authenticity and the zero-normalized cross-correlation (ZNCC) for structural preservation (Fig 2a). In general, the optimal value of $t_{start}$ is preferred to be set as early as possible to ensure the maximum structural correlation between sketches and image data. Additionally, since the generation process is iterative, selecting an earlier $t_{start}$ directly translates to lower generation times. We empirically determined $t_{start} = 400$ and $\sigma = 1$ to offer a good trade-off between generative capabilities and structural preservation. This is further supported by the observation that structural correlation is diminishing in regions of low contrast for $t_{start}$ higher than the determined value (Fig 2b). An additional benefit of synthetic data is indicated by the fact that slight inaccuracies of manual annotations were not present in simulated data, enabling the creation of error-free image-mask pairs even in presence of annotation errors. Moreover, feature representations of synthetic and real image data were shown to be similar for those settings (S1 Fig). All following experiments were conducted using the optimized settings.

## Correcting segmentation errors

While manually annotated structures were used during the optimization of the parameters to test the generative aspects of the pipeline without the influence of structural differences, using manually annotated data does not represent a practical scenario for generating fully-annotated image datasets. To assess different strategies for automation of the generative process, rough segmentations were employed as sources to obtain sketches for various organisms and cell

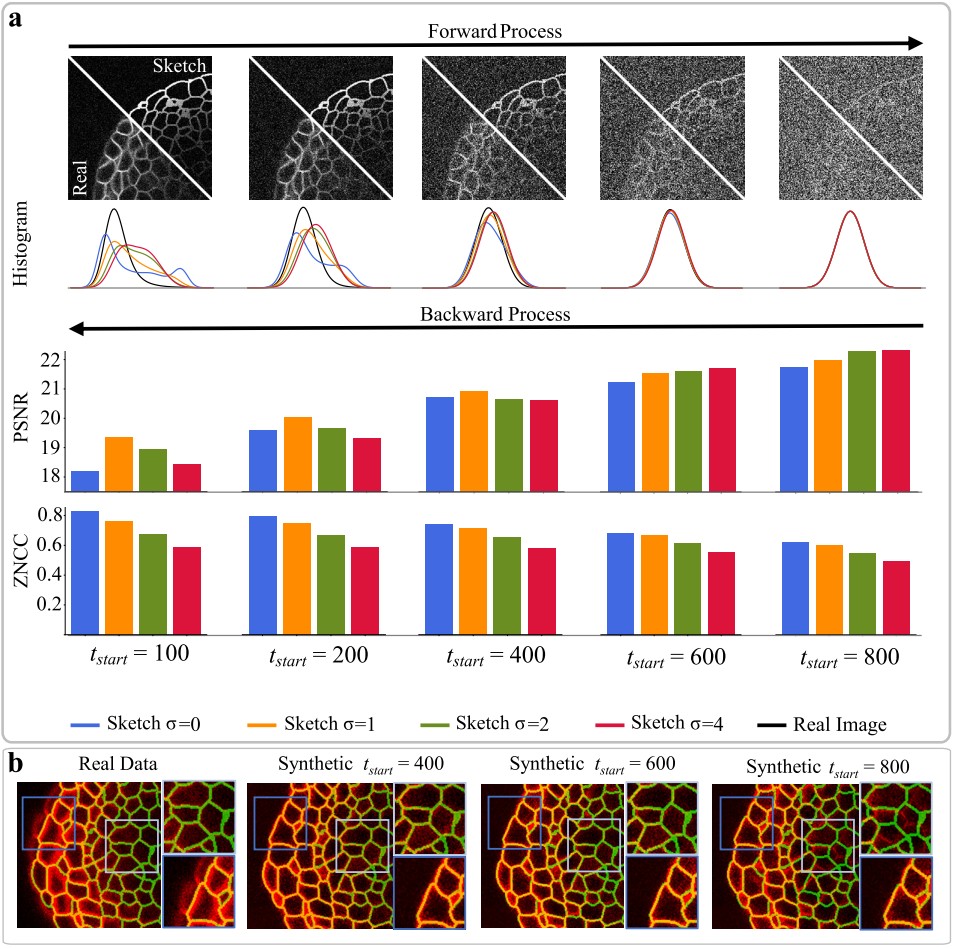

**Fig 2. Pipeline optimization.** (a) Noisy data created by the forward process from either real images or sketches needs to be sufficiently similar to allow for the generation of realistic image data in the backward process, assessed by histograms. For the backward process, peak signal-to-noise ratio (PSNR) and zero-normalized cross-correlation (ZNCC) are used as metrics, to assess the realism of image data generated from different starting points $t_{start}$ and sketch blurring factors $\sigma$. (b) Overlays of generated image data (red) and annotation masks (green) show how structural correlation is diminishing with increasing $t_{start}$ in regions of low contrast, while manual annotation inaccuracies even present in regions of high contrast do not appear in simulated data.

types. Publicly available generalist segmentation approaches [26, 27] and publicly available silver truth annotations [19] were used to obtain rough representations of cell shapes for nuclei in 3D *C. elegans* [19, 20], 3D *D. rerio* [21], 2 mouse stem cells [19, 22], 2D HeLa cells [19, 23, 24] and for cellular membranes in 3D *D. rerio* [21]. Image quality was assessed in regions where annotations were available using the PSNR as a metric, with mean scores ranging between 19.58 dB and 29.97 dB across all data sets (S2 and S3 Figs). Although the annotations may contain errors that could affect the reported quality scores, they do not have an impact on the application of the proposed pipeline, since the generated image data is directly correlated to the structures present in the annotations.

## Simulations for annotation-free segmentation

Relying on segmentations for collecting realistic structures can limit the scalability of the pipeline, as it is constrained by the availability of accurate generalist segmentation approaches. On

**Table 1. Forward process evaluation.** Bhattacharyya distance $D_B$ and Kullback-Leibler divergence $D_{KL}$ calculated for noisy samples generated by the forward process for different timesteps $t_{start}$. Sketches were used as a basis for the forward process, which were initially smoothed by Gaussian filtering with standard deviation $\sigma$. Highlighted values are obtained for data generated with the empirically determined optimal settings.

| $t_{start}$ | $\sigma$ | $D_B$ | $D_{KL}$ |
|---|---|---|---|
| 100 | 0 | 0.1261 | 0.3969 |
| | 1 | 0.0750 | 0.2398 |
| | 2 | 0.1274 | 0.4323 |
| | 3 | 0.1757 | 0.6214 |
| 200 | 0 | 0.0718 | 0.2276 |
| | 1 | 0.0402 | 0.1331 |
| | 2 | 0.0668 | 0.2366 |
| | 3 | 0.0911 | 0.3323 |
| 400 | 0 | 0.0103 | 0.0371 |
| | **1** | **0.0041** | **0.0153** |
| | 2 | 0.0099 | 0.0380 |
| | 3 | 0.0141 | 0.0546 |
| 600 | 0 | 0.0004 | 0.0016 |
| | 1 | 0.0001 | 0.0004 |
| | 2 | 0.0006 | 0.0022 |
| | 3 | 0.0010 | 0.0040 |
| 800 | 0 | 0.0000 | 0.0001 |
| | 1 | 0.0000 | 0.0002 |
| | 2 | 0.0001 | 0.0003 |
| | 3 | 0.0001 | 0.0004 |
| 1000 | 0 | 0.0001 | 0.0003 |
| | 1 | 0.0001 | 0.0005 |
| | 2 | 0.0001 | 0.0004 |
| | 3 | 0.0000 | 0.0001 |

the other hand, simulations present a more complex scenario but offer greater potential for generalizability and scalability. The main challenge lies in finding a simulation technique that can accurately reproduce the structural features visible in real image data, to help closing the domain gap between real and synthetic image data. Despite these challenges, conducting experiments with simulation approaches allows for exploring the full potential and limits of the pipeline. As directly assessing the realism of the image data generated from simulated sketches was challenging due to the absence of corresponding real image data, we followed a more practical way of evaluation. Instead, we focused on determining the usability of the generated data as training data by training the Cellpose approach [28] from scratch, followed by its application to real image data. By using the accuracy of the segmentation results as a proxy, the realism of the generated data is indirectly assessed. Additionally, segmentation results were compared to those obtained by the publicly available pretrained Cellpose model, which served as a baseline and a reference for models trained on a large, diverse and manually annotated image dataset. Simulations were obtained for five different datasets including cellular membranes in 3D *A. thaliana*, and nuclei in 3D *C. elegans* [19, 20], 3D *T. castaneum* [19], 2D mouse stem cells [19, 22] and 2D HeLa cells [19, 23] (Fig 3a). The results presented in Fig 3b and S8 Fig demonstrate that both models perform comparably well for each dataset, despite the fact that the models trained solely on synthetic image data were trained on a small dataset including only 200 generated samples. We intentionally chose a small synthetic dataset, such

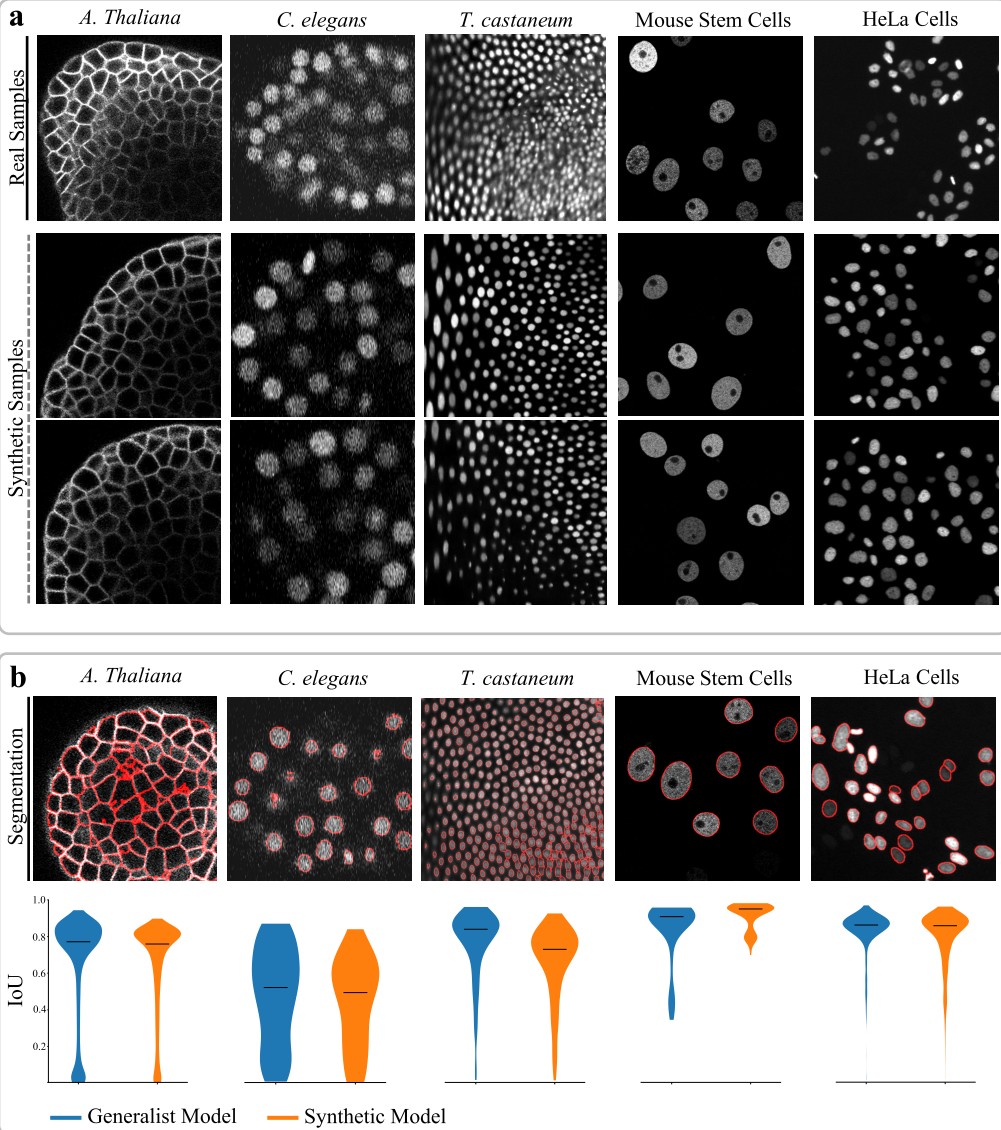

**Fig 3. Application Examples.** (a) Real image samples and fully-synthetic image samples generated by the diffusion model using simulated structures. (b) The Cellpose segmentation approach [28] is trained on synthetic datasets and applied to real image data to generate results (red overlay) without requiring human-generated annotations. Intersection-over-Union (IoU) scores obtained for a publicly available generalist model trained on a large collection of manually annotated image data (blue) and the model trained on synthetic data (orange) are shown as violin plots with indications of median values (black bar). All datasets are publicly available from [10, 18, 19, 21, 24].

that the results represent the quality rather than the diversity of the generated samples. With sparse ground truth data available for evaluation, a Wilcoxon rank-sum test [29] was conducted to identify potential differences in segmentation accuracy between the models. With the largest $p$-value reaching 0.0002 for the dataset showing mouse stem cells, the test confirmed that the performance of both models on all datasets is comparable. It is noteworthy that the model trained on synthetic data achieved comparable scores without the need for any human-generated annotations, surpassing the requirement for a large collection of annotated image data as in the generalist model. This capability enables the potential application of

segmentation models to entirely different datasets and structures, where generalist segmentation models would typically encounter challenges or limitations. To support those outcomes, the 3D Cellpose extension [27] was trained from scratch on real and synthetic image data from the same organism, presenting progress in narrowing the gap between real and synthetic domains, especially when compared to previous GAN-based methods [10] (S5 Fig). However, this demonstrates further room for improvement of the proposed approach, as the domain gap is still evident and the near perfect accuracy for an IoU threshold of 0.5 indicates the absence of highly challenging regions. Contrarily, this can be interpreted as evidence for the completeness of annotations provided for this data set, demonstrating one strength of generated datasets. Further experimental outcomes indicate that within a training configuration utilizing real image data, as much as 70–80% of the real image samples can be substituted by synthetic image samples while maintaining segmentation accuracy (S6 Fig).

However, there is one limitation to this approach, as the noise introduced during the forward process makes it challenging to produce very dark cells, with the structural information potentially getting lost in the added noise. Consequently, dimly illuminated regions pose difficulties for accurate segmentation, since the models trained on synthetic data may not be fully-equipped to handle all challenges posed by real data. This is supported by the decreased segmentation accuracy observed in datasets that provide manual annotations specifically for dimly illuminated regions (S7 Fig). Despite this limitation, the obtained segmentation scores demonstrate the capability of training specialized models that achieve state-of-the-art segmentation results in a fully-automated manner with data generated by an intuitive and unsupervised approach. Moreover, beyond those limitations, the presented approach offers substantial intuitive strength, enabling the generation of varied background illuminations and complex scenes featuring overlapping cells, both straightforwardly achieved through indications provided within the sketches (S4 Fig).

## Materials and methods

### Denoising diffusion probabilistic models

The concept of denoising diffusion probabilistic models (DDPM) used in the proposed pipeline employs a gradual noising process $q$, which is defined as a Markovian chain iteratively adding a small portion of noise to an image $\mathbf{x}_0$ until reaching pure noise $\mathbf{x}_T$ by following a cosine-based schedule $\beta_t$ [30]:

$$q(\mathbf{x}_t \mid \mathbf{x}_{t-1}) = \mathcal{N}(\mathbf{x}_t; \sqrt{1 - \beta_t}\mathbf{x}_{t-1}, \beta_t\mathbf{I}), \text{ with } t \in (0, T). \qquad (1)$$

For the data generation procedure, a corresponding backward process is defined in which a neural network is trained to iteratively reverse the forward process. Therefore, at every stage of the backward process, the model is tasked to predict the noise component $\epsilon_\theta$ introduced into $\mathbf{x}_0$ to transform it into $\mathbf{x}_t$. Subsequently, this prediction guides the acquisition of the preceding sample $\mathbf{x}_{t-1}$ by

$$x_{t-1} = \frac{1}{\sqrt{\alpha_t}}\left(\mathbf{x}_t - \frac{1 - \alpha_t}{\sqrt{1 - \alpha_t}}\epsilon_\theta(\mathbf{x}_t, t)\right) + \tilde{\beta}_t\mathbf{z}, \qquad (2)$$

where $\mathbf{z} \sim \mathcal{N}(0, \mathbf{I})$ is of the same size as $\mathbf{x}_t$ [14]. Repeated in an iterative fashion, this leads to the generation of realistic image data $\hat{\mathbf{x}}_0$ from $\mathbf{x}_T$.

## Network design

The underlying network architecture is a U-Net [31, 32] with pixel-shuffle upsampling [33] in the decoder path, and an additional conditional input in all blocks providing sinusoidal embeddings of the timepoint $t$ [14]. For all experiments, a maximum number of $T = 1000$ diffusion steps is used and the network is trained for 5000 epochs.

## Datasets

The utilized datasets show either cellular membranes or cell nuclei obtained from 2D(+t) and 3D fluorescence microscopy experiments. For all datasets, annotation masks are either obtained from manual annotations, automatically obtained unrefined segmentation or simulation approaches, to demonstrate use cases with various conditions. Multiple sophisticated approaches for automated simulation of cellular structures have already been proposed, ranging from physics-based methods [34], statistical shape-models [10, 11] and spherical harmonics [10, 35], to deep learning-based methods [12, 36, 37]. For simplicity, this work focuses on simulation approaches utilizing basic geometrical functions to create cellular structures. Generally, in cases where simulation approaches are used to generate cell nuclei, a foreground region is generated to roughly represent an organism outline or a region of interest, which is filled with cell nuclei at random locations. Each nucleus shape $\mathcal{R}$ at position $(x_{\text{center}}, y_{\text{center}}, z_{\text{center}})$ is simulated as an ellipsoid with radius $r_{\text{nuclei}}$ and directional scaling factors $(s_x, s_y, s_z)$ following

$$\mathcal{R}_{\text{ellipsoid}}(x, y, z) := \sqrt{\left(\frac{x - x_{\text{center}}}{s_x \cdot r_{\text{nuclei}}}\right)^2 + \left(\frac{y - y_{\text{center}}}{s_y \cdot r_{\text{nuclei}}}\right)^2 + \left(\frac{z - z_{\text{center}}}{s_z \cdot r_{\text{nuclei}}}\right)^2} \leq 1. \quad (3)$$

In case of 2D data, the z-dimension is omitted. Additionally, each nucleus is randomly rotated by angle $\alpha \in (0, 2\pi)$ around arbitrary axes and distorted to obtain more irregular shapes. Details for each data set are explained in the following paragraphs.

**Arabidopsis thaliana (3D).** A publicly available 3D fluorescence microscopy image data set showing the meristem of *A. thaliana* [18], including manually corrected annotation masks. Additionally, simulated annotation masks based on statistical shape models published in [10] are utilized for further experiments. The size of the synthetic image data averages to (511, 495, 221) voxel to mimic the image resolution of the real data set (we refer to [10] for more details). Since the real microscopy image data shows declining signal intensity towards the organism center, sketches are generated by linearly decreasing the simulated intensity signal towards the organism center accordingly. The availability of manually corrected and simulated annotation masks allows to use this data set for detailed experiments of the presented methods.

**Caenorhabditis elegans (3D).** A data set containing 3D image stacks of developing *C. elegans* [19, 20]. Low-quality automatically obtained silver truth annotations for all images and very sparse manually obtained ground truth annotations for a small selection of image slices are additionally provided. Annotations are simulated by outlining the foreground region as an ellipsoid located at the image center and filling the determined area with a variable amount of nuclei. The nuclei radii $r_{\text{nuclei}}$ dynamically decrease inversely proportional to the amount of nuclei in the region. To simulate cell morphology after mitosis, a random selection of 10% of cells are shrunken along two axis to form a more cylindrical shape. Sketches are formed by randomly choosing nuclei illumination. The shape of the simulated image is set to (512, 708, 35) voxel in correspondence to the real image resolution, and nuclei parameters are empirically set to $(s_x, s_y, s_z) = (1, \mathcal{U}(0.5, 1), 0.09)$, with $\mathcal{U}$ being a uniform distribution.

**Tribolium castaneum (3D).** A 3D data set showing nuclei of developing *T. castaneum* embryos [19]. Low-quality automatically obtained silver truth annotations for all images and very sparse manually obtained ground truth annotations for a small selection of image slices are additionally provided. Annotations are simulated by outlining the foreground region as a sphere located at the image center and nuclei are densely positioned at the outer boundary of the foreground region. Similar to the real data set, a cartographic projection is used to transform the 3D space into multiple stacked 2D projections of the organism surface. Therefore, the image space is considered in spherical coordinates $(r, \theta, \phi)$ originating at the image center and for a total of 13 subsequent fixed radii $r$ the spherical surface is mapped to a 2D space $(x, y) = (\theta, \phi)$, causing the poles to appear stretched. Nuclei illumination is randomly chosen to form the final sketch. The shape of the simulated image is set to (2450, 1700, 13) voxel in correspondence to the real image resolution, and nuclei parameters are empirically set to $r_{nuclei} \in (5, 6)$ and $(s_x, s_y, s_z) = (1, 1, 1)$.

**Danio rerio (3D).** A multi-channel 3D data set showing fluorescently labeled cell membranes and nuclei in two different zebrafish embryos [21]. Corresponding automatically obtained annotations published in [10] are used as a basis to create sketches. To impose a realistic intensity variance within the sketches, signal intensity linearly decays along the z direction. All images have a spatial size of (512, 512, z) voxel, while z is in the range of 104 to 120.

**Mouse stem cells (2D).** A 2D data set showing Mouse stem cells [19, 22]. Low-quality automatically obtained silver truth annotations for all images and very sparse manually obtained ground truth annotations for a small selection of images are additionally provided. For simulation of annotations, the whole image region is considered as region of interest and cells are placed at random positions, while avoiding overlaps. To introduce more irregular nuclei shapes, each nuclei is altered using a deformable transformation modeled with B-splines [38]. To form the final sketch, nuclei illumination is randomly chosen and barely illuminated nucleoli are simulated by randomly placing 0–2 small dark circles within each nuclei. The shape of the simulated image is set to (1024, 1024) pixel in correspondence to the real image resolution, and nuclei parameters are empirically set to $r_{nuclei} \in (30, 45)$ and $(s_x, s_y) = (\mathcal{U}(0.75, 1), 1)$, with $\mathcal{U}$ being a uniform distribution.

**HeLa cells (2D).** A 2D data set showing HeLa cells [19, 23]. Low-quality automatically obtained silver truth annotations for all images and very sparse manually obtained ground truth annotations for a small selection of images are additionally provided. Annotations are simulated by constructing regions of interest as randomly placed and overlapping circular regions within the image. The resulting foreground region is filled with nuclei that are additionally altered using a deformable transformation modeled based on B-splines [38]. Nuclei illumination is randomly chosen to form the final sketch. The shape of the simulated image is set to (700, 1100) pixel in correspondence to the real image resolution, and nuclei parameters are empirically set to $r_{nuclei} \in (10, 20)$ and $(s_x, s_y) = (\mathcal{U}(0.5, 1), 1)$, with $\mathcal{U}$ being a uniform distribution.

**HeLa cells (2D+t).** A 2D+t data set showing temporal mitotic progression of HeLa cells [24]. Each frame is centered on one single cell, which is tracked and manually annotated for a total of 90 frames each. For processing with the proposed pipeline, the temporal image stack is treated as a regular 3D image stack with a size of (96, 96, 90) voxel. Sketches are created by homogeneously setting the intensity of each annotated cell to the mean intensity identified within the annotated region of each respective real image frame.

**Cervical cells (2D).** A 2D data set consisting of a total of 945 cervical cytology images [39], showing cytoplasm and nuclei of overlapping cells. The images are split into two different

sets with 900 and 45 samples each and with an increasing amount of cell overlap. For the experiments conducted in this work, the larger split was used for training and the smaller split was used for testing. Sketches were created by homogeneously setting the intensity of each annotated cytoplasm and nuclei region to the mean intensity identified within the annotated region of each respective real image. Furthermore, overlapping regions were darkened by 10% per additional cell involved in the overlap to impose realistic image features.

## Metrics

For evaluation of synthetic image quality different metrics were used, which are listed and described in the following. $\mathcal{X}$ describes the set of all $n$ pixel or voxel positions within the image data and $P(\mathbf{x})$ is considered as discrete intensity distribution of an image $\mathbf{x}$.

- Bhattacharyya Distance:

$$D_B = -\ln\left(\sum \sqrt{P(\mathbf{x}_t^{\mathcal{I}}) \cdot P(\mathbf{x}_t^{\mathcal{M}})}\right). \tag{4}$$

- Kullback-Leibler Divergence:

$$D_{KL} = \sum P(\mathbf{x}_t^{\mathcal{I}})\log\left(\frac{P(\mathbf{x}_t^{\mathcal{I}})}{P(\mathbf{x}_t^{\mathcal{M}})}\right), \tag{5}$$

with $\mathcal{I}$ representing the image domain and $\mathcal{M}$ representing the mask domain.

- Peak Signal-to-Noise Ratio (PSNR):

$$PSNR = 20 \cdot \log_{10}\left(\frac{1}{\sqrt{MSE}}\right), \tag{6}$$

with

$$MSE = \frac{1}{n}\sum_{x \in \mathcal{X}} (\mathbf{x}_0 - \hat{\mathbf{x}}_0)^2. \tag{7}$$

- Zero-Normalized Cross-Correlation (ZNCC):

$$ZNCC = \frac{1}{n}\sum_{x \in \mathcal{X}} \left(\frac{\mathbf{x}_{0,x}^{\mathcal{M}} - \mu_{\mathcal{M}}}{\sqrt{\sigma_{\mathcal{M}}}} \cdot \frac{\mathbf{x}_{0,x}^{\mathcal{I}} - \mu_{\mathcal{I}}}{\sqrt{\sigma_{\mathcal{I}}}}\right), \tag{8}$$

with $\sigma$ representing the intensity variance and $\mu$ representing the mean intensity of data from the image domain $\mathcal{I}$ and the mask domain $\mathcal{M}$ respectively.

- Intersection over Union (IoU):

$$IoU = \frac{|y_{pred} \cap y_{gt}|}{|y_{pred} \cup y_{gt}|}, \tag{9}$$

with $y_{pred}$ representing the predicted segmentation mask and $y_{gt}$ representing the ground truth segmentation mask.

## Conclusion

Overall, the consistently high PSNR values for the synthetic image data and the segmentation results comparable to state-of-the-art approaches trained with large annotated datasets emphasize the realism of the generated data and demonstrate its practical usability. The presented pipeline tackles the issue of manual annotation demands in segmentation applications and proposes a shift towards identifying setups for acquiring sketches of cellular structures. This objective is usually more universal, simpler to address, and not influenced by the unique illumination and noise attributes of particular datasets. If effective methods for these sketches can be established, corresponding data can be synthesized and utilized as error-free training material for subsequent approaches, leading to fully-unsupervised and automated applications. We demonstrated that the generation of these sketches can be achieved through simulation approaches or by utilizing publicly available generalist segmentation methods. Inaccuracies or errors of those initial segmentations were eliminated in the generative process, leading to the acquisition of realistic error-free training data. Consequently, the application of deep learning-based segmentation approaches became more accessible for datasets with limited and absent annotations. To further contribute towards the goal of reaching annotation-free segmentation pipelines, all fully-annotated fully-synthetic image datasets are publicly available at https://osf. io/dnp65/, and code for training and application is available at https://github.com/stegmaierj/ DiffusionModelsForImageSynthesis.

## Supporting information

**S1 Fig. Latent feature representation.** 2D feature representations obtained with t-SNE from the latent representation of an autoencoder for real 3D *Arabidopsis thaliana* image data [18] and corresponding synthetic data generated from sketches of manual annotations. Additionally, feature representations obtained for raw sketches serve as a reference. Since this data set contains large-scale image data, each image stack is partitioned into patches to reduce computational demand. During application, a feature representation for each single patch is obtained and averaged to derive an overall feature description of the entire image stack. Results indicate that the diffusion model learns the average distribution of real image data, since latent representations of synthetic data is enclosed by representations obtained for real image data. Sketch representations form a more distinct cluster, further promoting the realism of synthetic image data.
(PDF)

**S2 Fig. Quantitative results of synthetic data.** PSNR values presented as boxplots, calculated between real image data and synthetic image data generated from corresponding silver truth and segmentation masks. Those silver truth segmentations are generated with automated approaches and validated to be reliable for training purposes. Whiskers range from the 5th to the 95th quantile, median values are indicated as orange line while mean values are depicted as green triangle and boxes represent the interquartile range. The involved datasets are 3D *Caenorhabditis elegans* (CE) [19], 2D Mouse Stem Cells (GOWT1) [19], 2D HeLa Cells (HeLa) [19], 3D Nuclei and Membranes of *Danio rerio* (DRNuc,DRMem) [21], 2D+t mitotic progression in Mouse Stem Cells (HeLa+t) [24] and 2D overlapping cervical cancer cells (Cerv) [39]. Using corresponding sketches and the optimized settings of $t_{start} = 400$ and $\sigma = 1$, the backward process was used to replicate corresponding image samples and PSNR values were calculated to assess similarity between synthetic and real versions. The *Danio rerio* multi-channel data and the temporal HeLa data present special cases, which demonstrate limitations of the

proposed approach.
(PDF)

**S3 Fig. Qualitative results of synthetic data.** Examples of real image data and corresponding synthetic image data generated from corresponding silver truth and segmentation masks. The involved data sets are publicly available in [10, 19, 20, 22–24]. In case of the temporal HeLa data set, the 2D+t image stacks were treated as regular 3D images to allow for the generation of temporally consistent textures within the cellular and background regions. This was motivated by observations from previous experiments showing that a frame by frame generation of the temporal data is feasible but prone to the generation of slight textural inconsistencies that we were not yet able to prevent otherwise.
(PDF)

**S4 Fig. Special cases of generated image data.** Examples of real image data and corresponding synthetic image data generated from corresponding annotations of overlapping cells [39] (top). Moreover, examples of varying background illumination in *C. elegans* [19, 20] was generated by adding indications within the sketches (bottom). This demonstrates the intuitive strength of the proposed approach, as more complex scenes of overlapping cells can be realistically generated and position-dependent texture characteristics can be straightforwardly imposed and controlled.
(PDF)

**S5 Fig. Segmentation accuracy on real and synthetic image data.** 3D Cellpose [27] segmentation models are trained on synthetic image data and real image data respectively, using the 3D *Arabidopsis thaliana* data set [18]. Both models are tested on real and synthetic image data. A model trained on GAN-generated image data is tested on real image data for a comparison between diffusion-based and GAN-based approaches. IoU scores are calculated for each cell in the image data and an IoU threshold provides the basis to formulate an accuracy as the ratio of precise segmentations to the total quantity of cells. It should be noted that no augmentation or dedicated pipeline tweaking was used during the training of the segmentation approaches, in order to obtain results that purely reflect the capabilities of the synthetic data.
(PDF)

**S6 Fig. Segmentation accuracy for different ratios of real and synthetic image data.** To present an analysis from a practical point of view, 3D Cellpose [27] segmentation models are trained on datasets containing a mix of synthetic and real image data from the 3D *Arabidopsis thaliana* data set [18]. The total quantity of training images was kept consistent throughout all experiments and all models are tested on real image data with the segmentation accuracy being reported by the IoU metric. Whiskers of the boxplots range from the 5th to the 95th quantile, median values are indicated as orange line while mean values are depicted as green triangle and boxes represent the interquartile range.
(PDF)

**S7 Fig. Segmentation accuracy on further datasets.** Solid lines show results obtained for segmentation models solely trained on synthetic data evaluated on the sparse manually annotated ground truth. Dotted lines show determined accuracies when considering the silver truth annotations as predictions (not provided for *T. castaneum*), and comparing results against the ground truth. Data splits, ground truth and silver truth were provided by the Cell Tracking Challenge [19]. Note that manual annotation are only provided for a small fraction of cells visible within the image data, and annotated cells often focus the most challenging regions, which

are typically difficult to generate.
(PDF)

**S8 Fig. Qualitative examples of segmentation results.** Examples of sparse ground truth annotations (top) and corresponding segmentations obtained by models solely trained on fully-synthetic image data generated from simulated sketches (bottom). The datasets and ground truth annotations are provided by the Cell Tracking Challenge [19]. Note that the annotations have never been used for training and are merely depicted for qualitative comparison. No data augmentation was applied.
(PDF)

## Author Contributions

**Conceptualization:** Dennis Eschweiler, Ina Laube.

**Data curation:** Dennis Eschweiler, Rijo Roy, Abin Jose.

**Funding acquisition:** Johannes Stegmaier.

**Methodology:** Dennis Eschweiler, Rüveyda Yilmaz.

**Project administration:** Johannes Stegmaier.

**Resources:** Daniel Brückner, Johannes Stegmaier.

**Software:** Dennis Eschweiler, Rüveyda Yilmaz, Matisse Baumann, Daniel Brückner.

**Supervision:** Daniel Brückner, Johannes Stegmaier.

**Validation:** Dennis Eschweiler.

**Visualization:** Dennis Eschweiler, Rüveyda Yilmaz.

**Writing – original draft:** Dennis Eschweiler.

**Writing – review & editing:** Rüveyda Yilmaz, Rijo Roy, Abin Jose, Johannes Stegmaier.

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
