## [Decision Letter · Decision Letter 0]

17 Oct 2023

Dear Mr. Eschweiler,

Thank you very much for submitting your manuscript "Denoising diffusion probabilistic models for generation of realistic fully-annotated microscopy image datasets" for consideration at PLOS Computational Biology.

As with all papers reviewed by the journal, your manuscript was reviewed by members of the editorial board and by several independent reviewers. In light of the reviews (below this email), we would like to invite the resubmission of a significantly-revised version that takes into account the reviewers' comments.

Dear Authors,

Your paper has completed the review period. The reviewers found your work to be of interest and well written, however have expressed a number of concerns that should be addressed prior to publication. Please see especially the major concerns raised by Reviewer 2 as well as the comments about the 3D and 2D+t visualization and validation raised by Reviewer 1. I am thus suggesting a Major Revision and Resubmit for this manuscript.

Best,

-Adam

We cannot make any decision about publication until we have seen the revised manuscript and your response to the reviewers' comments. Your revised manuscript is also likely to be sent to reviewers for further evaluation.

Sincerely,

Adam Charles

Guest Editor

PLOS Computational Biology

Daniel Beard

Section Editor

PLOS Computational Biology

Dear Authors,

Your paper has completed the review period. The reviewers found your work to be of interest and well written, however have expressed a number of concerns that should be addressed prior to publication. Please see especially the major concerns raised by Reviewer 2 as well as the comments about the 3D and 2D+t visualization and validation raised by Reviewer 1. I am thus suggesting a Major Revision and Resubmit for this manuscript.

Best,

-Adam

Reviewer's Responses to Questions

**Comments to the Authors:**

Reviewer #1: Accept with minor revisions - see weaknesses section

Summary and contributions

The authors have developed an approach using denoising diffusion probabilistic models trained on real image data and annotations to generate synthetic datasets with built in annotations. These synthetic datasets are then used to augment training of a segmentation model that may be used to segment additional real, unannotated datasets.

Strengths

The authors make use of a modern generative technique to create high-quality synthetic microscopy image datasets. This approach is innovative and has several advantages:

1. Use to flexibly generate images with annotations over a range of different types of datasets

2. Can be tailored to augment specific datasets for annotation training

3. Can be used to validate a variety of segmentation approaches with synthetic ground-truth data

Weaknesses

The work as shown is somewhat limited in its application. The primary limitation comes from the description of the sample datasets as 2D+t or 3D datasets but a lack of treatment of these datasets as such. The ability of the approach to generate useful synthetic data for training annotations relies in part on its ability to faithfully reproduce the artifacts and abberations that are common to microscopy datasets.

1. The apparent requirement of the annotation be presented as a set of non-overlapping regions in a 2D image limits the ability to generate . As seen in Fig 2b or 3a, the real data close-up shows features of structures that exist in 3D that are partially overlapping in the 2D slice.

2. The diffusion model does not appear capable of generate more general artifacts over different regions of the image. This is most clearly seen in Fig 3a T. castaneum where the real dataset has different noise properties over different regions but not in the synthetic datasets

3. The examples provided do not include examples of particularly challenging datasets, such as densely packed and labeled cells with close borders. The focus on labeled membranes or nuclei is understandable but it would be useful to better see limitations of the technique with cytoplasmic labeling.

4. It would be instructive to see synthetic datasets generated explicitly on 2D+t data and see if motion artifacts can be generated in data with moving cells.

Correctness/Clarity

The approach is sound. The study is well-written and clearly describes the method well with useful examples and several metrics for assessing the work. Some additional analyses/datasets would be useful to address the weaknesses describe above.

Code and datasets are available online

Prior work

The authors reasonably cited prior literature in the field.

Reproducibility

Datasets and code are available online.

Reviewer #2: This paper proposes the use of diffusion models (DDPM) for the generation of fully annotated synthetic 2D/3D fluorescence microscopy images that can be used to train segmentation models. Specifically the authors describe domain specific pipelines to synthetically create annotation masks from which rough "images sketches" are produced that then are fed into a domain specific DDPM (starting at a intermediate scheudle step) to produce the corresponding images. They demonstrate that a cellpose model trained on these synthetic datasets performs comparable to a generalist model.

In general the paper is well written and addresses the important problem of annotation scarcity in microscopy instance segmentation, especially in the case of 3D segmentation where ground truth annotation of training data is a major bottleneck in practice. I only have two major issues regarding 1) the utility for segmentation models, and 2) the feasibility of mask/sketch generation for new domains, both of which I think need to be discussed/addressed.

1)

- It is shown for a given domain that training on synthetic data and applying to real data leads to results comparable to the generalist cellpose model (which was trained on a variety of different domains and is actually a 2D model). To actually demonstrate the claimed main utility of the approach (elevating the annotation burden), however, one would need to compare it to a model that was solely trained on the real annotations from the *same* domain. The supplement figure 4 seems to show the results of that experiment, but the legend is unclear (Is this "SynDiff2Real" vs "Real2Real"?). If so, the reduced accuracy (~ 0.87 for synth vs ~0.95 for real for thresh 0.5) actually would show that there is still quite a large drop potentially due to the domain gap? I think adding these numbers (for the real domain specific 3D cellpose model) to the main Fig3 is merited.

- Related to that, it would be important from a practical point of view to know at which amount of real annotation masks this real domain specific model surpassing the one trained on synthetic data. This should be a fairly simple experiment and would inform the reader what order of magnitude of real annotations would be needed to "catch up" with the (cheap) synthetic annotations.

2) The mask/sketch creation process seems to be very intricate and domain specific. First, one needs a domain specific generative model of the mask (e.g. nuclei) distribution and then another for the domain specific mask->sketch generation (e.g. decreasing intensity along the z axis for A. thalania or the nucleoli placement for Mouse stem cells). It thus seems that the needed amount of knowledge how to design this pipeline for a new (and potentially more complicated) domain might be overly challenging thus defeating the premise of the paper. Is there a way to assess how important the used pipeline details are for the final segmentation results? (e.g. does it drop considerable when not using the axial intensity profile and the shape model of S[3] or is it actually quite robust to it?)

Minor:

- What are the memory requirements esp. for the 3D image generation? What image size is feasible to generate?

- Could you add the rough training times/GPU specs needed for the 2D and 3D DDPM?

- The way how to use the code on new datasets is currently hard to understand from the github repo description. (How to convert the images? How to adapt the parameters, typos like "utils.h5_conveter.prepapre_masks"). I would suggest to provide example images/masks to download together with the concrete script calls to run the training/prediction pipeline on those.

**Have the authors made all data and (if applicable) computational code underlying the findings in their manuscript fully available?**

Reviewer #1: Yes

Reviewer #2: Yes

PLOS authors have the option to publish the peer review history of their article (what does this mean?). If published, this will include your full peer review and any attached files.

Reviewer #1: No

Reviewer #2: No
---

## [Decision Letter · Decision Letter 1]

5 Feb 2024

Dear Mr. Eschweiler,

We are pleased to inform you that your manuscript 'Denoising diffusion probabilistic models for generation of realistic fully-annotated microscopy image datasets' has been provisionally accepted for publication in PLOS Computational Biology.

Best regards,

Adam Charles

Guest Editor

PLOS Computational Biology

Daniel Beard

Section Editor

PLOS Computational Biology

Reviewer's Responses to Questions

**Comments to the Authors:**

Reviewer #1: I thank the authors for their responses to my concerns and I believe the revised manuscript is suitable for publishing as is.

Reviewer #2: The authors addressed all of my issues and improved the code repository, so I am happy to recommend publication.

**Have the authors made all data and (if applicable) computational code underlying the findings in their manuscript fully available?**

Reviewer #1: Yes

Reviewer #2: Yes

PLOS authors have the option to publish the peer review history of their article (what does this mean?). If published, this will include your full peer review and any attached files.

Reviewer #1: No

Reviewer #2: No

---

## [Editor Report · Acceptance letter]

13 Feb 2024

PCOMPBIOL-D-23-01470R1 

Denoising diffusion probabilistic models for generation of realistic fully-annotated microscopy image datasets

Dear Dr Eschweiler,

I am pleased to inform you that your manuscript has been formally accepted for publication in PLOS Computational Biology. Your manuscript is now with our production department and you will be notified of the publication date in due course.

With kind regards,

Zsofia Freund
